# Genomic Instability Evolutionary Footprints on Human Health: Driving Forces or Side Effects?

**DOI:** 10.3390/ijms241411437

**Published:** 2023-07-14

**Authors:** Laura Veschetti, Mirko Treccani, Elisa De Tomi, Giovanni Malerba

**Affiliations:** GM Lab, Department of Neurosciences, Biomedicine and Movement Sciences, University of Verona, 37134 Verona, Italy; laura.veschetti@univr.it (L.V.); mirko.treccani@univr.it (M.T.); elisa.detomi@univr.it (E.D.T.)

**Keywords:** genomic instability, DNA repair, human complex disorder, evolutionary genetics, ncRNA, aging, neurodegenerative diseases

## Abstract

In this work, we propose a comprehensive perspective on genomic instability comprising not only the accumulation of mutations but also telomeric shortening, epigenetic alterations and other mechanisms that could contribute to genomic information conservation or corruption. First, we present mechanisms playing a role in genomic instability across the kingdoms of life. Then, we explore the impact of genomic instability on the human being across its evolutionary history and on present-day human health, with a particular focus on aging and complex disorders. Finally, we discuss the role of non-coding RNAs, highlighting future approaches for a better living and an expanded healthy lifespan.

## 1. Introduction

A struggle of the collective drive toward complexity, auto-organization and genomic diversification against the “self” need to protect one’s own genomic information has been taking place since the beginning of life. On one plate of the scale, there is the dynamism of mechanisms involving genomic variability: gene transfer, duplication, rearrangements, recombination and exchange of mobile genetic elements, which are the events that most likely lead the drive to biological complexification [1,2]. On the other plate, there is the reliability of DNA conservation: a plethora of repair systems adapted and evolved together with organisms’ genomes to preserve them from corruption during individuals’ and cells’ lifespans and reproduction [3].

In this work, we will discuss the evolutionary significance of genomic (in)stability—intended as the propensity of the genome to change or to stay unchanged—and its resonance on human health. Since DNA repair systems have been extensively described in a recent comprehensive and well-detailed review by Basu and Essigmann [4], rather than covering the molecular aspects of such mechanisms, we provide an evolutionary overview of genomic instability in different organisms. In particular, we aim to explore the mechanisms involved in genomic stability across the tree of life and then focus on genomic instability significance in humans and its involvement in aging and complex disorders (CDs). Finally, we explore the role of non-coding RNAs (ncRNAs) and highlight open fields of biomedical research.

## 2. The Balance between Variability and Conservation

Genomic structural integrity and functional stability are constantly threatened by DNA and chromatin damaging agents ranging from exogenous sources, such as environmental toxins (e.g., polycyclic aromatic hydrocarbons), ultraviolet light (UV), ionizing radiation and mutagenic chemicals, to endogenous processes, including DNA replication and repair errors, epigenetic dysregulation, telomere shortening, spontaneous decay of DNA, transposable elements (TEs) insertions and oxidative stress (e.g., generation of reactive oxygen species) [4] (Figure 1). The consequences of damaging events are displayed as a wide variety of genomic wounds which comprehend base mismatches, single-strand breaks (SSB), double-strand breaks (DSB), inter-strand crosslinks, intra-strand crosslinks, bulky adducts and genomic rearrangements [5]. It has been almost a century since researchers started studying DNA damage, way before the discovery of the DNA double-helix structure [6]: the earliest works focused on UV radiation effects [7] and the genetic mechanisms involved in healing from these lesions [8]. Since its very beginning, this field became immediately of great importance and interest from different biological perspectives, and numerous studies arose to deeply investigate the causes and consequences of DNA damage, as well as the physiological responses.

In order to remediate DNA damage, a plethora of DNA repair mechanisms (Figure 2) have emerged across the tree of life, and their importance is evidenced by the presence of redundant, complementary and conserved repair systems. Indeed, such systems are so important that in the debated research for a minimal genome, researchers found that up to 5% of the required genes have to be committed to DNA repair mechanisms [9,10]. The rationale behind having such a wealth of repair systems lies in the fact that each mechanism is able to recognize and fix specific damage substrates [11]. For example, base-excision repair (BER) can resolve base mismatches, single-strand breaks, and intra-strand crosslinks by generating an apurinic/apyrimidinic site (AP site), which is then cleaved by an AP endonuclease, thus creating a single-strand break that is then closed by nucleotide synthesis [12]. In addition, base mismatches can also be detected and corrected by mismatch repair (MMR) together with replication slippages [13], whereas single-strand breaks are also repaired by SSB repair (SSBR) [14]. Conversely, DSB—the most deleterious form of DNA damage—can be either resolved by homologous recombination (HR), which uses a homologous DNA template for repair [15], or by nonhomologous end-joining (NHEJ) through DNA ends ligation [16]. Finally, inter-strand crosslinks can be processed via the Fanconi anaemia (FA) pathway and nucleotide excision repair (NER), which removes bulky, helix-distorting lesions, including intra-strand crosslinks [17].

We should remember that both DNA damage and repair systems leave genomic scars typical of the mechanisms involved, thus generating genomic diversity. Such scars can either be driving forces to physiological processes—as adaptive immune response and meiosis (e.g., DSB-mediated recombination) [18,19]—or side effects of a defective restoration of information. For example, MMR deficiency may induce single nucleotide substitution and variation in the length of short repetitive DNA sequences (e.g., microsatellites) [20], whereas HR defects typically lead to loss-of-heterozygosity, allelic imbalances extending to the telomeres and large-scale rearrangements [21]. For this reason, the remains of genomic scars are precious footprints that guide scientists through the investigation of adaptation and evolution of life by offering a window toward the past that might allow a better understanding of the present and, for us humans, of the increased risk for diseases.

DNA damage and repair, change and conservation of information, and adaptation and selection are factors whose weights reside on opposite plates of the scale, and the balance between them is ruled by laws that still need to be fully understood. In this context, the environment emerges as a not-so-hidden judge favouring one plate over the other. A notable example is the onset of “hypermutator” phenotypes in response to environmental factors [22,23]. This phenomenon implies increased mutation rates and can be observed in several events, such as microbial adaptation and cancer evolution [24,25,26]. Many theories have been proposed to explain genomic instability in terms of fitness in certain environments. Focusing on microbial adaptation, the prevailing hypothesis states that genomic instability is a profitable event because it increases the microbial population’s overall chance of survival [27,28,29]. Breivik and colleagues proposed a shift of perspective from colony to single-cell level by focusing on the biological cost of DNA repair: genomic instability arises because DNA repair may cost more than the errors it prevents in mutagenic environments [30]. Even though the debate on the possible advantages brought by genomic instability is still open, unstable genomes seem to be transiently favoured in stressful environments, whereas stable ones adapt more successfully in the long run [31,32]. Therefore, we may argue that DNA instability offers the possibility to gain adaptive advantages by assaying many solutions in a short time, but it cannot persist for a long time due to the intrinsic inability (by definition) to reach a stable, profitable status.

## 3. The Guardians of Genomic Stability across the Tree of Life

One of the first attempts to study the evolution of repair systems was carried out by Aravind, Walker and Koonin in 1999 [11]. The scientists searched for homologues of the repair proteins sequences of model organisms *Escherichia coli* and *Saccharomyces cerevisiae* in many bacteria, archaea and eukaryotes genomes. The authors reported a considerable heterogeneity in repair systems across the tree of life: they found that proteins involved in DNA repair seem to follow the “domain Lego” principle, according to which proteins are generated by copying, shuffling and recombining a limited number of conserved domains. Moreover, horizontal gene transfer—between bacteria and archaea, as well as between organellar and eukaryotic genomes—was proposed as a key mechanism contributing to the richness of repair systems. The fundamental bricks of the repair system complex were already present in common ancestors, and therefore, the study of the first forms of life could offer precious insights regarding the guardians of genomic stability across the tree of life and their evolution.

The last universal common ancestor is supposed to have evolved in a high temperature and anoxic environment when the Earth magnetic field was still weak and the planet vulnerable to ionizing radiation. Since that time, the environment and conditions in which life develops and thrives have dramatically transformed and the revolutionary change has been the increase in oxygenation levels, determined by the evolving organisms themselves [33]. Indeed, before the Great Oxidation Event (around 2.4–2.0 giga-annum ago), the predominant threats to genomic integrity were base loss, cytosine deamination, and damages induced by UV light, ionizing radiation, and alkylation, whereas in today’s world oxidative DNA damage is a substantially bigger concern [34]. Recently, Prorok and colleagues investigated the role of atmospheric oxidation on the evolution of DNA repair pathways and found that it had great impact on the spring out of novel repair mechanisms [35]. The authors proposed that ancient forms of life had simple, efficient and accurate mechanisms of direct damage repair (DR) and nucleotide incision repair (NIR), which operated without excision and de novo DNA synthesis. These systems were activated according to the type of damage: AP sites and deaminated residues were remediated though AP endonucleases in NIR pathway, whereas UV and alkylation damages were resolved by DNA glycosylases in DR. One of the first alternatives to DR and NIR were photolyases, which are nearly universally conserved in bacterial taxa and intervene in case of damage generated by UV light [35]. Shortly after, the inclusion of an additional base excision step into NIR gave rise to the modern BER pathway, which is one of the four universal DNA repair systems together with NER, MMR and HR [36]. However, such mechanisms proved to be insufficient when faced with the increase in frequency and spectrum of oxidative DNA lesions generated by the change in cellular metabolism of complex organisms (e.g., appearance of oxidative phosphorylation), thus new pathways emerged in bacteria, archaea and eukaryotes independently (Figure 3) [35].

### 3.1. Nucleotide Excision Repair Pathway

Modern NER evolved only after the separation of bacteria and eukaryote domains and introduced the ability to repair a wide range of bulky helix-distorting DNA adducts by damage recognition, lesion excision and DNA synthesis [37]. In particular, NER is present in bacteria in the form of the widely conserved UvrABC protein complex: UvrA is involved in damage recognition, UvrB is a helicase that opens the dsDNA and UvrC a nuclease that operates cuts on both sides of the damage [38]. In eukaryotes, an analogous and more complex NER pathway can be found: XPC-hr23b performs damage recognition, transcription factor IIH opens the dsDNA and binds to XPA and RPA proteins, and XPF-ERCC1 and XPG nucleases are finally recruited to perform lesion excision [38]. Interestingly, it has been proposed that the NER pathway in eukaryotes seemingly descended from bacteriophage anti-restriction ArdC proteins fused with an archaeal-derived papain-like peptidase domain [36,39]. Only a few archaeal taxa harbour UvrABC systems, likely acquired via horizontal transfer from bacteria [36]. In archaea, a lack of dedicated NER machinery has been reported, and a variety of alternative mechanisms to repair bulky lesions can be found [37]. For instance, Grogan suggested the possibility that bulky DNA lesions could alternatively be removed by pathways that restart stalled replication forks in UvrABC-deficient archaea [40]. In humans, NER impairment has been linked to a variety of conditions like xeroderma pigmentosum, trichothiodystrophy and Cockayne syndromes [41].

### 3.2. Mismatch Repair System

Another universal repair pathway is MMR which has as key players MutS and MutL proteins in bacterial species (except for *Actinobacteria*) and their respective homologues in eukaryotes [42]. Interestingly, most archaea lack *MutS* and *MutL* genes homologues and the few groups that harbour them tend to be temperature mesophiles such as halophiles and methanogens, which most likely acquired these genes via the horizontal transfer. In the great majority of archaeal organisms an alternative pathway, named endonuclease mismatch specific (EndoMS), that detects and corrects mismatches can be found [43]. Curiously, EndoMS is also present in bacterial genomes belonging to the *Actinobacteria* phylum, where MutS and MutL are absent [44]. MMR deficiency in humans is associated with microsatellite instability (MSI) across different cancer types as colorectal and endometrial carcinomas [45].

Overall, endonucleases containing iron-sulfur clusters appear to have a relevant role as sensors of oxidative DNA damage in a wide variety of organisms [46,47]. Among them: endonuclease III (Nth)—which excises oxidized pyrimidines—is conserved in archaea; endonuclease V (Nfi/EndoV) homologs are present in all domains of life and show activity towards DNA and RNA substrates suggesting a role in RNA editing [48,49]; UV damage endonuclease, which is found only in fungi (e.g., *Schizosaccharomyces pombe*, *Neurospora crassa*), bacteria and in *Sulfolobaceae*, *Methanomicrobia* and *Halobacteria* archaeal lineages [50,51,52]; and, finally, endonuclease Q family which is found in some bacteria (*Bacillus pumilus* and *B. subtilis*), participating in antiviral defence mechanisms [53].

### 3.3. Double-Strand Break Repair

Both HR and NHEJ systems are involved in the repair of DSBs. HR is one of the universal DNA repair systems and is implicated in the restart of DNA replication at stalled forks [54]. It is also involved in promoting genetic diversity via DNA transfer [55,56]. However, this is an energetically demanding and complex process, and, for this reason, simpler but less accurate pathways, such as NHEJ, operate alongside HR. In eukaryotic cells, NHEJ is commonly used in the G1 phase of the cell cycle since it does not depend on the presence of a homologous DNA duplex [38]. The NHEJ pathway has also been reported in bacteria by Weller and colleagues, who underlined the need of a DNA end-binding bacterial Ku protein for the correct operation of this system [57]. Due to the rarity of Ku protein homologues in archaea, NHEJ is only found in a small number of species and the microhomology-mediated end joining (MMEJ) system could be a valuable alternative for double-strand break repair in these organisms [58]. In human beings, DSB repair inefficiency has been related to Nijmegen breakage syndrome, ataxia telangiectagia-like disorder and Seckel syndrome [59].

### 3.4. DNA Repair: Going Viral

At the end of this excursus through the branches of the tree of life, it is necessary to include a reflection on viruses. Rather than entering the debate on whether they are living beings or not, we want to focus on their role in DNA repair mechanisms activation. Indeed, DNA damage response can be activated by incoming viral DNA, during the integration of retroviruses, in response to aberrant DNA structures generated upon active viral DNA synthesis, or during persistence of extrachromosomal viral genomes [60,61]. In addition, some viral proteins are able to trigger the production of reactive oxygen species (ROS) that can cause oxidative DNA damage, thus activating repair pathways. Damage response may be either beneficial or detrimental to viruses. Since viral DNA genomes are subject to damaging agents, DNA repair mechanisms can become advantageous to them as quality control tools to proofread viral genomes prior to assembly. For this reason, the downstream signalling pathway that leads to apoptosis or senescence in eukaryotes is often mitigated by specific viral proteins to ensure cell survival during viral replication [62]. On the other hand, repair systems could intervene on viral genomes thus impairing their replication. Subsequently, viruses evolved molecular mechanisms such as virus-encoded direct and indirect antagonists of damage repair pathways that prevent the suppression of viral replication [60]. Such mechanisms can prove to be dangerous in humans when viral DNA integration drives the overexpression of oncoproteins, inhibition of DNA damage checkpoints, or structural rearrangements leading to genomic instability [61].

## 4. Genomic (in)Stability in *Homo sapiens*: Just a Matter of Luck?

Going on through the tree of life, we now set out to focus on the present-day humans. As all living organisms on Earth, humans are exposed daily to a multitude of endogenous and exogenous sources of DNA damage [63]. DNA lesions are far more frequent than is commonly thought. AP sites—which can arise spontaneously, be triggered by ROS, or are determined by the BER mechanism—are a great example to understand how frequent DNA damages occur in everyday life: it has been estimated that approximately 50,000 to 200,000 AP sites arise daily in every mammalian cell [64] and this phenomenon seems to take place at a higher rate in the elderly [65]. The SSB represent another recurrent DNA damage event in human cells: Martin and Liu estimated that, in mammalian cells, DNA SSBs account approximately for 20,000 to 40,000 daily damages, but that their number may dramatically increase depending on lifestyle, diet and habits [66].

Mechanisms of DNA damage and repair have been deeply investigated to explain the consequences of damages on health, to investigate the possibility to restore physiological status of damaged cells, and to assess their association with disorders and pathological conditions. Numerous studies, brilliantly overviewed by Schumacher and colleagues [67], have investigated these processes from several perspectives, pointing out a tight relation between molecular [68], cellular [69], and systemic [70] levels. Specifically, at the molecular level, DNA damage can lead to genomic instability, telomeric dysfunction, epigenetic alterations, deregulation of transcription patterns, proteostatic stress and mitochondrial dysfunction; at the cellular level, stem cell exhaustion and cellular senescence; systemic repercussions regard complex processes as signalling, inflammation and sensing. All these are the main events summarizing the landscape of DNA damage in humans.

### 4.1. Nuclear and Mitochondrial DNA

Nuclear DNA and mitochondrial (mtDNA) are two separated genomes, indeed they present structurally different DNA molecules: the diploid linear nuclear genome and the multi-copy haploid circular mitochondrial genome. In recent decades, several studies pointed out their interconnection relatively to the handling of DNA damage [71,72,73]. For example, Baulch reported how genomic instability induced by radiation may alter cellular epigenetic mechanisms and can reduce mitochondrial functions; at the same time, mitochondrial dysfunction hampers the cell epigenetic profiles [74]. Moreover, the relationship between nuclear and mitochondrial instability seems to be involved in the aging process (i.e., the multifactorial biological processes of declining physiological functions) and in common pathologies, such as cancer and neurodegenerative disorders: suggestive examples were reported on breast cancer [75], Alzheimer’s disease [76] and Huntington’s disease [77].

Several DNA repair mechanisms emerged throughout evolution, and more than 125 genes involved in such systems have been identified in humans [78,79]. It has been observed that the mutation rate is higher in mitochondria than in the nuclear DNA [80,81]. Specifically, mtDNA is more prone to oxidative damage due to the presence of a higher concentration of ROS—which account for approximately 10,000 daily DNA lesions per cell [66]—and the lack of chromatin protection [82]. The accumulation of mtDNA damage can lead to mitochondrial dysfunction and has been linked to age-related diseases, such as Parkinson’s disease [83] or Werner syndrome [84], and to different types of cancer [85].

Little is known about mtDNA repair pathways, with great discrepancies between well-known and verified processes (such as BER) and hypothesized or lightly demonstrated mechanisms (such as MMR or HR), as emphasized by Rong and colleagues [82] and Patel et al. [86]. Not all the repair mechanisms present in the nuclei are also present in the mitochondrial genome (Figure 4); thus, mitochondria make use of nuclear genes and transcription products to repair occurring DNA lesions [87]. Researchers all agree that the primary repair process in mitochondria is the BER mechanism [88]. Mitochondrial BER takes advantage of several glycosylases that are encoded by nuclear genes, although they are present at lower levels when compared to the nucleus and, in some cases, in different spliced isoforms [89]: 8-oxyguanine DNA glycosylase-1 (OGG1) isoform beta (homolog of the nuclear OGG1 isoform alpha), Nth-like 1 (NTHL1), Nei-like 1 (NEIL1) and Nei-like 2 (NEIL2), alkyladenine DNA glycosylase (AAG), MutY glycosylase homolog (MUTYH) and uracil N-glycosylase (UNG) isoform 1 (homolog of the nuclear UNG isoform 2). Moreover, a unique component in the mitochondrial BER mechanism has been identified in the enzyme DNA polymerase γ (POLG) as first reported by Longley and colleagues [90] and later confirmed by Tahbaz et al. [91]. On the contrary, the NER mechanism does not seem to be present in the mitochondrion. A recent study by Karikkineth and colleagues showed the presence of some traces of excision repair cross-complementing proteins in the mitochondria of patients affected by Cockayne syndrome [92], but due to the absence of several NER key components, this repair mechanism is unlikely to happen in the mitochondria in the same way as it is known in the nucleus [81]. Similarly, the mitochondrial MMR mechanism seems to differ quite a bit from its nuclear equivalent: if in the nucleus, the MMR process is well-characterized, in the mitochondrion, this mechanism is still not well understood [93]. The MMR nuclear actors, respectively, MutS homolog (MSH) and MutL homolog (MLH), are not present in the mitochondria [93]; however, the MMR seems to take place in the mitochondrial environment [94], and its activity seems to depend on the Y-box binding protein 1 (YB-1) [95]. Furthermore, little is known about the processes involved in SSBR and DSBR. SSBR machinery is likely to be present in the mitochondria due to the presence of different proteins involved in this process and shared with the mitochondrial BER pathway, but conflicting interpretations on the role of poly-ADP-ribose polymerase 1 (PARP1)—a key element in the identification of SSB—must be resolved to reach clear evidence on this repair mechanism [96]. As regards DSBR, researchers have identified the presence and the activity of BRCA1 [97] and 53BP1 [98] proteins in mammalian mitochondria, but their role in damage repair and maintenance of genomic stability is far from clear.

### 4.2. Evolutionary Insights

Genomic instability is gradually acquiring a central role in our knowledge on human evolution, providing novel insights on our past as humankind, as well as new perspectives on future therapeutical targets. Breivik and Gaudernack [30] were probably the first to hypothesize the thin trade-off intrinsic to genomic instability: the loss of genomic stability might give to evolutionary mechanisms the opportunity to take action and to explore possibilities for fitness advancement and novel adaptation [99], but, at the same time, it might be indirectly associated with an increased risk of several late-onset diseases [100], which from an evolutionary perspective are unfavourable explorations. From a similar perspective, Little gave insight on the concept of randomness, profoundly inborn in evolution and, hence, in the evolutionary trade-off concept [101]: not only does evolution take place in random ways but also DNA damage and genomic instability happen randomly either in time (in terms of lifetime as well as triggering causes) or in space (in terms of genomic location or cellular localisation). Furthermore, the decrease in genomic stability seems to be related to a dramatic increase in the mutation rates, which in turn may be related to newer opportunities for evolution and to potential triggers of various diseases, as in the case of exposure to ionizing radiation, as experimented and reported by Morgan [102] and Nagar and Morgan [103]. Indeed, Baird discussed genomic instability as the mean to generate genetic diversity in different populations, setting the ideal conditions in which evolution can perform at its best by providing great genomic changes in short evolutionary time periods [104].

When it comes to humans, this novel perspective amplifies the research focus not only on present-day biological processes but also on our genomic history as human beings: all our genetic changes are written in our DNA, hence investigating DNA lesions scarred in our genomes might provide novel insights on host-pathogen co-evolution, genomic instability, and disease onset and/or severity. To better understand genomic (in)stability across the human evolution, Cordaux and Batzer [105] focused on the analysis of transposable elements (TEs) as major players still influencing the human genome functionality. TEs are DNA sequences that can move within the genome, originally identified in maize [106] and subsequently confirmed in humans [107]. Nearly half of the human genome seems to be composed of TEs, and this could be an underestimation due to the presence of extremely ancient TEs which are no longer recognisable [108]. In human, several TEs have been identified such as DNA transposons, long terminal repeats (LTRs) retrotransposons as the human endogenous retroviruses (HERVs), non-LTR retrotransposons as long interspersed element 1 (LINE-1 or L1), Alu and SINE-R-VNTR-Alu (SVA—composed of short interspersed nuclear element of retroviral origin [SINE-R], variable number of tandem repeats [VNTR], and Alu), and other minor elements [109]. All these elements are densely distributed along the genome and have a strong impact in shaping human genomic structural and functional features, carrying information of past adaptation as well as the seeds of evolution, either in terms of fitness improvement and genomic innovation [110], or as causes of genomic instability and genetic disorders [111]. However, TEs are currently not mobile in the human genome and their last activity is definitely far from present days. TEs were active at different ages along the evolutionary history of mammalian organisms, and their origin is dated several millions of years (Myr) ago (Figure 5).

The oldest among TEs are the L1 elements [112], which are thought to have been active since 150 Myr ago: these elements represent roughly 17% of the human genome size and, despite their great number of copies (more than 500,000), less than 100 seem to be currently functional [113] and several of them seem to be implicated in age-related diseases [114] and cancer [115]. Around 65 Myr ago [116], Alu elements started their mobile activity, resulting in more than 1 million copies in today’s human genome. Despite being the most abundant TE, Alu elements do not seem to be functional and are usually defined as parasitic DNA [117]. However, their relevance seems to be related to the human evolution and development, since their activity and integration has originated from a common ancestor belonging to the superorder of the *Euarchontoglires* (also defined as Superprimates), to which primates belong [118]. With the early primates, around 37 Myr ago [119], DNA transposons interacted with genomes of archaic hominids and other mammals [108], enhancing their distinctive ‘jumps’ [120] and integration into genomic sites [121]. DNA transposons account for nearly 3% of the present-day human genome and are implicated in different RNA pathways, including piRNA biogenesis [122].

Getting closer to the evolution of hominids and the advent of *Homo sapiens*, the mobility and integration of human endogenous retroviruses and of SINE-R-VNTR-Alu took place, respectively, around 30 Myr [123] and 25 Myr ago [124]. These two elements show various similarities in terms of nucleotide sequences and genetic features. HERVs account for nearly 8% to 10% of mammalian genomes [125]. Even though their current function is not fully understood, they seem to be implicated in several inflammatory and immune disorders [126], like multiple sclerosis [127] and systemic lupus erythematosus [128] and cancer [129].

SVA elements [130] are not so present in the human genome, accounting for roughly 3000 copies [131], probably due to their nonautonomous nature and their LINE-1-related origin [132]; however, like most of the previous reported TEs, SVA seems to be associated with inflammatory conditions and autoimmune diseases, such as amyotrophic lateral sclerosis [133], systemic lupus erythematosus and Chron’s disease [134]. TEs showed to be fundamental for hominoid and human evolution, shaping their development and genomic advancement for several hundreds of millions of years and resulting in an increase in size of the human genome and a significant inter-individual variability in TEs content [135], turning out to be a highly informative vault for human evolutionary history. Moreover, their presence in present-day humans is a strong signal of evolutionary advantage, since they have been maintained in our genome for several millennia: some examples are TEs contribution to genetic innovation, such as the introduction of new genes in the whole known lifespan of humankind [105] and their implication in some regulatory networks [136], including immunity [137] and embryonic development [138]. However, the presence of TEs, which can vary within individuals of the same species, can be considered as one of the major causes threatening human health; moreover, integrated TEs are subjected to lesions and mutations similarly to the surrounding genomic regions. Particularly, TE polymorphisms can determine the deregulation of TEs activity that in turn lead to numerous diseases [139,140]: mitochondrial diseases [141], respiratory diseases [142] and several age-related diseases [143].

## 5. Genomic Instability, Aging and Late Onset Complex Diseases

In this paper, we undertake the arduous endeavour of discussing genomic instability. This topic is frequently addressed exclusively in relation to its contribution to different types of cancer. However, in this work, we present genomic instability as a process which encompasses not only DNA damage [67] but also mobile elements insertion, telomeric and microsatellite instability, mitochondrial dysfunction and epigenetic alterations. Genomic instability is a threat constantly affecting all living organisms, and, in humans, this phenomenon is particularly studied due to its implication in aging and its tight link to CDs and cancer [144]. Indeed, in the course of life, the human genome is subjected to a massive number of damages [64]: the great majority are fixed efficiently [63], whereas some of them escape the repair process and accumulate in the genome, impacting several processes and aging [67,145,146,147].

There is little evidence of association between DNA repair improvement and lifetime expansion [148,149], thus, indicating that such mechanism seems to have evolved to maintain DNA stability—and therefore health—only until reproductive age, without any regard for the fate of the individual in old age, both in terms of quality and length of life. Although life expectancy has markedly increased over the past century and a half, this was not matched with an extension of the healthy lifespan [150,151,152]. Since the aging process cannot be delayed, we are destined to spend more years of our life in old age, thus, allowing genomic instability by-products—that have not been subjected to natural selection—to play an unexpected role (e.g., late onset CDs) in this timeframe.

What is currently emerging from the scientific literature points at a tight link among genomic instability, aging and CDs: genomic instability has been proposed as an hallmark of the aging process, which is in turn one of the major risk factors to a variety of pathological conditions, such as functioning loss onset (i.e., the decrement in physical and/or cognitive functioning), chronic disease progression and increased infectious diseases susceptibility [153]. Therefore, from the biological and evolutionary perspectives, aging might be interpreted and investigated as a multitude of genetic complex traits [154,155]. Some phenomena such as telomeric and microsatellites instability and mitochondrial dysfunction emerged to be key factors involved in both aging and in the development of a variety of common diseases.

### 5.1. Telomeric Instability

Human telomeres are composed of a long stretch (up to tens of kilo-base pairs) of TTAGGG nucleotide repeats located at the end of each chromosome to protect them from degradation and ensure their stability [156,157]. Indeed, cells carry a variety of mechanisms and proteins—including the shelterin complex and telomerase—responsible for the maintenance of telomeres length [157]. However, the mitotic process determines a shortening of telomeres in daughter cells compared to the parent cell, thus telomeres have been proposed as “molecular clocks” for aging [158]. Moreover, shortened telomeres trigger replicative senescence and impair the regenerative capacity of tissues, which is undesirable in the case of pluripotent stem cells and adult stem cell compartments [159].

Impairment of telomeric maintenance and accelerated telomere shortening have been found to be associated with some of the leading causes of disease and death; among them: central obesity [160,161], lifetime accumulation of stress [162,163], increased risk of cardiovascular events [164,165], and reduced immune response to influenza vaccination [166]. In particular, somatic mutations in genes involved in telomeres maintenance have been linked to the functional decline of B lymphocytes, skeletal muscle cells, and neurons [158]. Additionally, germline mutations affecting such pathway have been reported to have a pivotal role in disease onset: genetic mutations associated with short telomeres have been shown to cause Hoyeraal–Hreidarsson syndrome, dyskeratosis congenita, pulmonary fibrosis, aplastic anaemia, liver fibrosis and several other severe medical conditions defined as “telomere syndromes” or telomeropathies [158,167]. Even though such diseases display a high level of phenotypic heterogeneity, comprising age of onset and severity of clinical manifestations, on the molecular level, they are all characterised by the presence of short telomeres, which have also been linked to gene expression alterations through the “telomere position effects” [168].

### 5.2. Microsatellites Structural Maintenance

Microsatellites are short tandemly repeated sequence motifs consisting of 1–6 bp that are typically repeated up to 50 times in millions of locations across the genome [169]. At least two main mechanisms can play a role in the failure of microsatellite structure maintenance: (1) DNA replication errors (e.g., due to polymerase slippage) that impact the length of microsatellites, and (2) defects in DNA repair mechanisms that determine an accumulation of errors leading to the generation of shorter/longer novel fragments (also known as MSI).

In the first case, the number of repeat units changes from one generation to the next due to replication slippage. In particular, alleles with a higher repeat number appear to be less stable than those with a lower number of repeats, which explains why a highly significant excess (compared to the expectation under the assumption of random effect) of long microsatellites has been observed in humans and across different species [170]. This type of instability can affect different genomic locations with a varying magnitude, which is reflected both in repeat expansion disorder onset timeframe (i.e., the greater the damage, the earlier the onset age, also known as anticipation) and phenotypic severity (i.e., ranging from mild to severe phenotypes). These findings suggest that there might be even loci carrying expanded repeats which manifest either a cumulative or peculiar effect in the elderly life.

MSI has been associated with defects of the MMR system [171], and has been linked to genomic instability in cancers [172,173], inflammatory diseases, such as Crohn’s disease [174] and Behçet’s syndrome [175], and Lynch syndrome [176]. Interestingly, MSI has recently been reported as also being involved in infertility by Wieland and colleagues, who proposed the presence of MSI as a biomarker of underlying DNA repair deficiencies resulting in idiopathic infertility [177].

### 5.3. Mitochondrial Dysfunction

Mitochondria are additional key players involved in aging and CD development. Actually, the role of mitochondria in aging is so determining that a “mitochondrial theory of aging” has been proposed [178]: with age, mitochondria accumulate ROS-induced damage and become dysfunctional, and the function of cells declines causing aging. It is apparent that mitochondrial dysfunction particularly affects organs that require high levels of energy such as the heart, skeletal muscles and brain [179]. Several reports indicate that mitochondrial dysfunctions may be involved in the development of different neuropsychiatric, neurodegenerative and developmental disorders given their key role in energy metabolism and neural apoptosis [180], among them: schizophrenia [181], Alzheimer’s dementia [182], bipolar disorder [183], autism spectrum disorder [184], attention deficit-hyperactivity disorder [185] and Parkinson’s disease [186]. Once more, the link between genomic instability, aging and CDs emerges and raises the question: can the aging process be studied as a complex trait, and can we intervene on its biological and environmental underlying mechanisms to expand our healthy lifespan?

### 5.4. Human Networking: The Systemic Complexity of Life

When highlighting the link between genomic instability, aging and CD, we must not forget that we—as humans—do not live as single entities but we are a part of complex systems and communities (Figure 6). In the past, the great majority of people lived in isolated groups composed of a small number of individuals (i.e., 50–100 people), and only recently—for evolutionary times—have people started gathering in larger cities. Such transition went hand in hand with a radical change in social dynamics that had repercussions on the population genetic scale: in the past, the genetic variation pool of individual communities was very limited and selective pressures (e.g., disease, famine, etc.) were particularly high, whereas today humans make up a very extensive community with an overall rich pool of genetic variation. Moreover, through scientific and technological advance our species was able to prevent the impact of selective pressure (e.g., vaccination, drugs, agriculture), thus modifying the genetic flow throughout generations by maintaining variants that in a natural setup would have been filtered out (e.g., due to random genetic drift, natural selection), even through background selection. This goes against the logic of natural selection, at least in the short term; hence, common traits that show an increasing prevalence might find their roots in many rare functional genetic factors under mild or no selective pressures.

Up to the XX century infectious diseases were the main cause of mortality, whereas, in recent decades, this role has been taken up by diseases such as cancers, cardiovascular diseases, and metabolic disorders [187]. This might indicate that the attenuation of selective pressures acting on human beings might allow variants with mild effects to play a detectable role in the long term, either through damage accumulation (e.g., threshold effect) or simply because they have the chance to manifest their effects with the progression of aging.

Shifting the focus from the “inside” to the “outside”, we should recall the “not-so-hidden judge”: the environment, considered as a complex system including classical environmental factors (e.g., pollutants) and inter-species dynamics among humans and other living organisms. Interestingly, recent research has shown that many conditions and CDs are influenced by both human and environmental microbiota composition. Among many findings, we report some notable ones: decrease in microbiological diversity in the everyday living environment has been proven to lead to immune tolerance dysregulation and has been proposed as one of the core reasons for the epidemic of immune-mediated diseases in western urban populations (i.e., hygiene hypothesis) [188]; intestinal microbiota has been shown to be associated with metabolic disorders as obesity and type 2 diabetes [189,190]; naturally diverse airborne environmental microbial exposures seem to modulate the gut microbiome and might provide anxiolytic benefits [191]; and dysbiosis—defined as an imbalance—of gut microbiota was present in stroke and transient ischemic attack patients [192], and has been identified as a possible major contributor to the elevated incidence of multiple age-related pathologies [193].

Such findings highlight once more the interconnectedness that runs among humans, all other living organisms and the environment, and urge scientists to adopt a one-health approach to unravel not only the current relationship among these actors, but also the ones involving our ancestors. Indeed, investigating such ancient equilibrium might allow us to gain insights concerning contemporary lifestyles’ consequences on health and promote healthy interactions with nature or dietary practices that might help us contrast many of the contemporary world diseases.

## 6. The Epigenome: Shedding Light on the Dark Side of the Genome

As emerges from this broad overview, the long-term survival of a species is naturally linked to adaptation and depends on a thin balance between genome stability and its intrinsic tendency to corrupt and change. Over the past decade, numerous studies have tried to identify classes of molecular mechanisms related to aging and disease. López-Otín and colleagues proposed a total of nine hallmarks, including the epigenome, that has emerged as an important player in the decline of cell function observed both in aging and late-onset CDs [194,195]. The epigenome consists of chemical alterations to the DNA and histone proteins that results in changes to the structure of chromatin and function of the genome that can be inherited from parent to offspring [196]. Moreover, it is considered to be responsible for DNA stability and gene expression in different tissues, thus influencing the phenotype variability. Indeed, epigenomic changes in humans (and across the tree of life) impact on CDs onset and aging (and on the different patterns of embryonic development) [197,198].

Functional studies in humans and model organisms have shown that epigenetic modifications are crucial at all stages of development because of their ability to regulate genes transcriptionally. Particularly, multiple epigenetic events were found altered across different species during aging: accumulation of histone variants, changes in chromatin accessibility, loss of histones and heterochromatin, histone modifications, and deregulated expression/activity of microRNAs (miRNAs) [199,200]. Over the years, aging has been associated with increased transcriptional noise characterized by aberrant production and maturation of both many mRNAs and ncRNAs [201,202]. With the advent of new sequencing technologies, several tissue- and organism-specific transcriptional signatures of aging have been identified [203,204,205]. Barth and colleagues have identified conserved aging-related transcriptional signatures that characterize all tissues of long-lived individuals [206,207]. These transcriptional signatures involve the downregulation of a specific class of miRNAs associated with aging, called geromiR, which can influence lifespan by negatively controlling the gene expression of target components that are part of longevity networks [194]. The first geromiR was identified by Boehm et al., in 2005, who reported that the loss of function of miR lin-4 in *Caenorhabditis elegans* mutants was associated with reduced lifespan compared to wild-type organisms, while the overexpression of miR lin-4 extended their lifespan [208]. Similarly, in the following years, other studies demonstrated that some miRNAs could promote longevity in *C. elegans*, whereas others showed a pro-aging effect [209]. Owing to the evolutionarily conserved nature of some of these miRNAs, it was reported that their the regulatory role likely extends to humans as well [195]. However, the class of geromiRs is not the only one implicated in aging and associated diseases.

### 6.1. The Non-Coding Impact on Coding

MicroRNAs are involved in the regulation of almost all cellular processes through specific downregulation of gene expression at the post-transcriptional level. Indeed, they can influence the translation of more than 60% of the protein-coding genes [210]. In addition to their intracellular functions, miRNAs can act as active messengers that trigger a systemic response. Among these, the group called inflamma-miRs can affect inflammatory pathways [211]. An excess of inflammatory activation has been associated with the development of major age-related diseases, such as cardiovascular disease, Alzheimer’s disease, rheumatoid arthritis, type 2 diabetes mellitus and cancers [212]. The dysregulation of most circulating inflamma-miRs may contribute to the development and progression of these diseases by cooperatively regulating a given biological process [213].

Although miRNAs have been well studied in humans, they are just the tip of the iceberg. A series of ncRNAs can play significant roles, among them: small nuclear RNAs (snoRNAs), circular RNAs (circRNAs), PIWI-interacting RNAs (piRNAs), and a large group of long non-coding RNAs (lncRNAs), including non-coding transcripts from intergenic regions (lincRNAs). These ncRNAs function as part of a complex network that intervenes in many processes, including aging and senescence, through the modulation of gene expression, genomic imprinting and nuclear organization [195,214,215]. Moreover, several studies have shown that ncRNAs play a crucial role in regulating genes involved in DNA damage repair mechanisms, and in maintaining genomic stability through the activation of cell cycle checkpoints and induction of apoptosis when the damage is irreparable [216]. In response to damage, the action of ncRNAs functions as a key node connecting the rapid DR-mediated protein modifications and the late response mediated by transcriptional regulation [217]. However, at the same time, DNA damage can alter ncRNA expression at multiple levels, including transcriptional and post-transcriptional regulation and degradation [213,218,219]. Alterations of their regulatory functions are particularly relevant in the context of aging.

### 6.2. The Diamond in the “Junk”

Unlike DNA mutations, epigenetic alterations and deregulations of ncRNAs—which were once considered “junk”—are theoretically reversible, offering opportunities for the development of new perspectives and insights on possible new therapeutic interventions [194,220]. In recent years, there has been growing interest in using ncRNAs as therapeutic agents for a wide range of pathologies. However, there are several challenges in designing effective therapies that exploit the effects of ncRNAs because multiple molecular mechanisms are involved in different pathologies. For example, it is essential to identify the best ncRNA targets or sets of targets for each pathological condition, prevent toxic and off-target effects and ensure the effectiveness of the delivery system and treatment stability [221]. Despite these challenges, several types of ncRNAs are currently undergoing clinical trials for the treatment of various diseases. For example, for miRNA-based therapies, some candidates in clinical trials include cobomarsen, which targets miR-155 for the treatment of blood cancers, and MRG110, a specific antagomir for miR92 for tissue repair [222]. Another interesting example of the application of ncRNA-based strategies in therapy is the use of coding circRNAs as a vaccine against SARS-CoV2. This has been developed to create an RNA vaccine that is thermally stable in both naked and encapsulated forms [222]. Given the gene expression regulatory role of ncRNAs in CDs and aging, and the recent report on ncRNA-targeted therapeutic interventions in cancer and infectious diseases, it is very likely that in the near future, novel ncRNAs-based personalized medicine tools will also be deployed in the context of aging and several common diseases.

## 7. Towards a Better Living

In this paper, we discussed some causes contributing to genomic instability which in turn represents one actor that emerged from a far past and now shapes human health. Particularly, we highlighted the tight relationship among genomic instability, aging and CDs, emphasising the role of ncRNAs as a possible new weapon to contrast the effect of the natural corruption of genomic information. Nonetheless, many challenges still need to be faced. Novel molecular approaches and ecological complex longitudinal study designs have to be employed to explore how genomic instability mechanisms impact human lives: among them, the pathways underlying DNA repair in mitochondria, the impact of lifestyle and environment, and the contribution of rare variants. The new knowledge will certainly contribute to the development of future approaches to better live in synergy with the environment, leading to an expanded healthy lifespan.

## Figures and Tables

**Figure 1 ijms-24-11437-f001:**
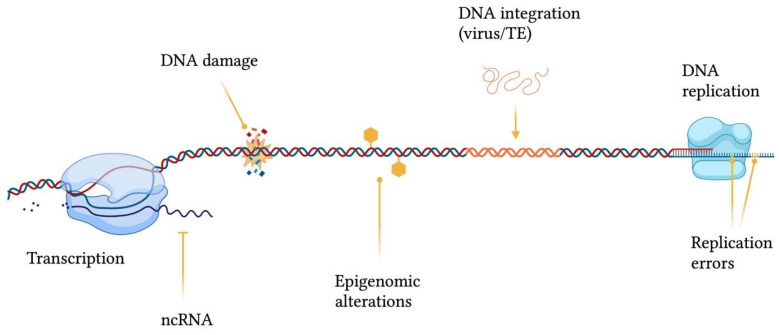
Sources contributing to genomic instability. ncRNA = non-coding RNA; TE = transposable element. This figure was generated using BioRender.

**Figure 2 ijms-24-11437-f002:**
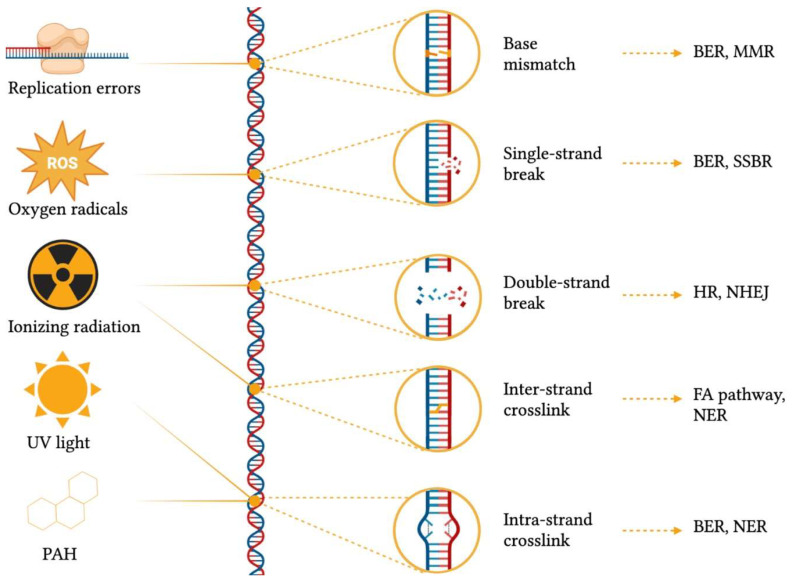
Common DNA damaging agents, types of genomic scars caused by different damage sources, and damage repair mechanisms. BER = base-excision repair; FA = Fanconi anaemia; HR = homologous recombination; MMR = mismatch repair; NER = nucleotide excision repair; NHEJ = nonhomologous end-joining; PAH = polycyclic aromatic hydrocarbons, ROS = reactive oxygen species; SSBR = single-strand break repair; UV = ultra-violet. This figure was generated using BioRender.

**Figure 3 ijms-24-11437-f003:**
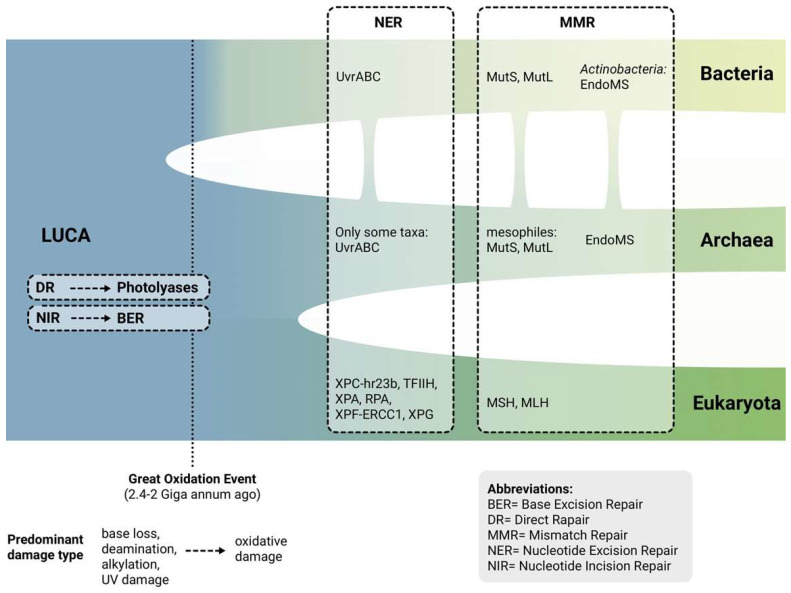
Schematic representation of the evolution of repair systems across the tree of life from the last universal common ancestor (LUCA) to today’s domains of life. Vertical branches indicate possible horizontal transfer of genes involved in the repair mechanisms.

**Figure 4 ijms-24-11437-f004:**
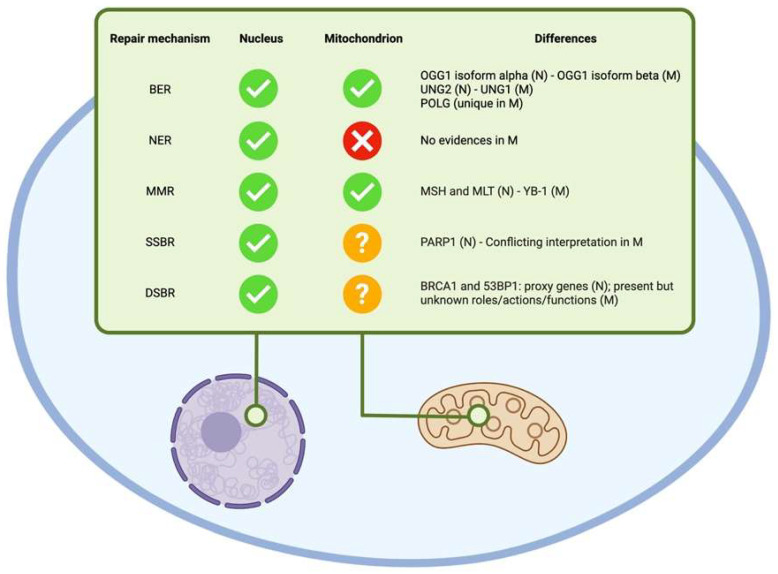
Nuclear and mitochondrial DNA repair mechanisms. BER: Base Excision Repair; DSBR: Double Strand Break Repair; MMR: Mismatch Repair; NER: Nucleotide Excision Repair; SSBR: Single Strand Break Repair; N: Nucleus/Nuclear; M: Mitochondrion/Mitochondrial. This figure was generated using BioRender.

**Figure 5 ijms-24-11437-f005:**
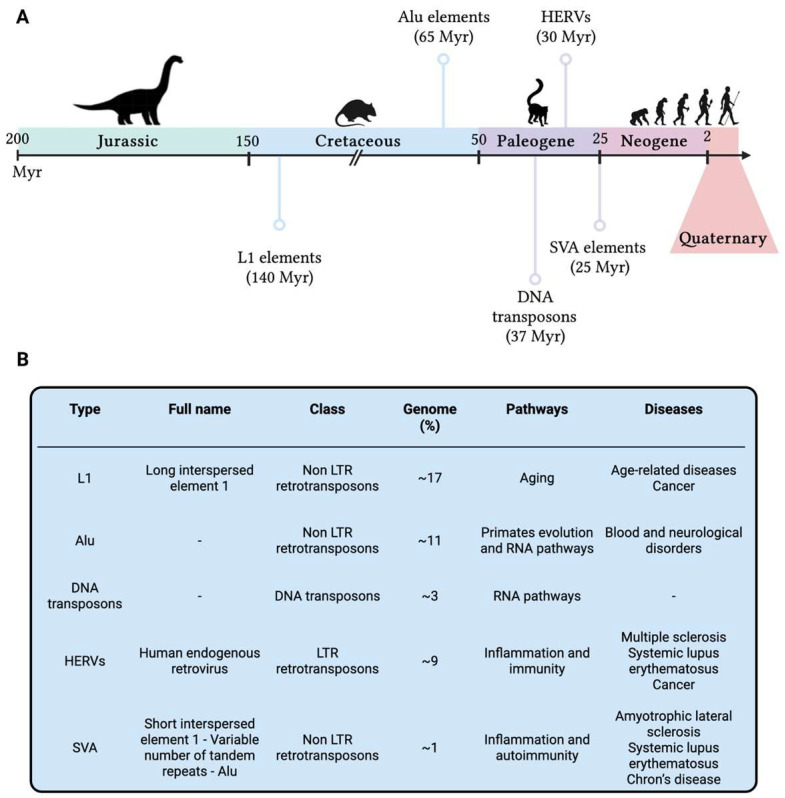
(**A**) Timeline of the activities of the primary transposable elements which had a fundamental role in the evolution of present-day humans. (**B**) Main transposable elements in the human genome reported in chronological order of activity. Genome = the percentage of human genome identified as TE type; Pathways = pathways in which TEs have been shown to play a role; Diseases = diseases in which TEs have been reported to be implicated. HERVs = Human Endogenous Retroviruses; L1 = Long Interspersed Nuclear Elements 1; LTR = Long Terminal Repeats; Myr = millions of years; SVA = SINE-R-VNTR-Alu. This figure was generated using BioRender.

**Figure 6 ijms-24-11437-f006:**
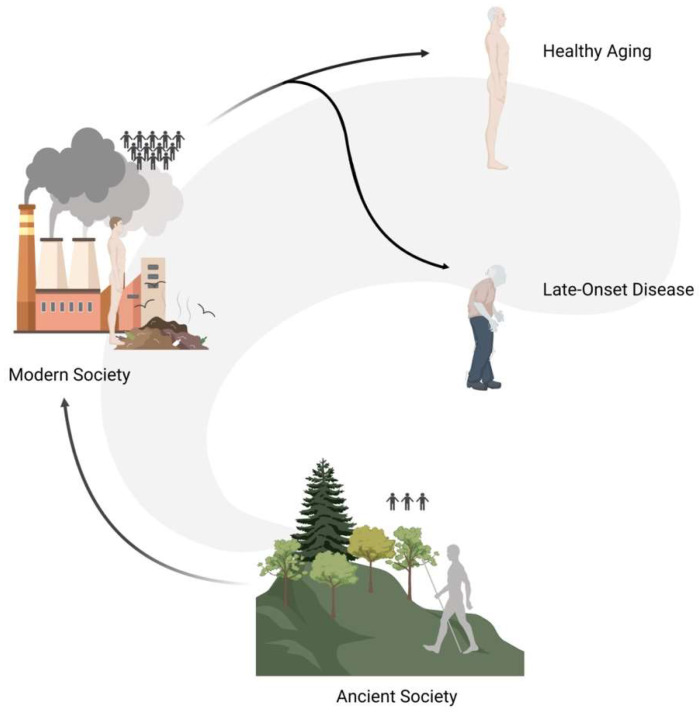
Graphical representation of social dynamics changes that had repercussions on the populations genetic level. In the past, people lived in isolated groups composed of a small number of individuals (ancient society), and only recently have people started gathering in larger cities (modern society). The modern ability to prevent the impact of selective pressures determined the maintenance of variants that in a natural setup would have been filtered out, thus possibly causing late-onset diseases.

## Data Availability

Not applicable.

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
