# Peer review of "Genomic Instability Evolutionary Footprints on Human Health: Driving Forces or Side Effects?"

_ijms, 2023, doi:10.3390/ijms241411437_

Round 1

Reviewer 1 Report

Title: Genomic evolutionary insights toward a long and healthy life

The present title looks very rudimentary and its review article so it should be catchy. I recommend  author can plan a title which could end with question marks.

Abstract: the language is too complicated to infer a clear-cut conclusion. So, recommend author to make it simple and try to rewritten the whole section in simple and small sentences.

The older references o be replaced with newer one preferably 80 % beyond 2018. 

Before start any content there must be an introduction which may includes the scope of article and impact of studies and t future direction. So, introduction need to be inserted.

I strongly recommend author to put a representative picture for each subheading under section for better under standing

 The guardians of genomic stability across the tree of life

Language must be improved

Here must be more section like limitation of genomic evolutionary process the epilogue, Future prospects

language and gramatical mistakes section rearrangement 

Author Response

Comment 1:

Title: The present title looks very rudimentary and its review article so it should be catchy. I recommend author can plan a title which could end with question marks.

Author response:

The title has been updated to: “Genomic instability evolutionary footprints on human health: driving forces or side effects?”

Comment 2:

Abstract: the language is too complicated to infer a clear-cut conclusion. So, recommend author to make it simple and try to rewritten the whole section in simple and small sentences.

Author response:

The entire manuscript underwent proof-reading and language has been improved.

Comment 3:

The older references o be replaced with newer preferably 80% beyond 2018.

Author response:

We thank the reviewer for this suggestion. However, we think that references to works published before 2018 are fundamental for this manuscript. Indeed, we not only set out to critically summarize recent advances in the field, but also aim to provide a comprehensive overview (by highlighting historic scientific breakthrough) of genomic instability in different organisms.

Comment 4:

Before start any content there must be an introduction which may includes the scope of article and impact of studies and t future direction. So introduction need to be inserted.

Author response:

A general introduction has been added to the manuscript.

Comment 5:

I strongly recommend author to put a representative picture for each subheading under section for better under standing

The guardians of genomic stability across the tree of life

Author response:

New figures have been added to the manuscript to clarify the concepts illustrated in each chapter and to better represent the novelty of our work.

Comment 6:

Language must be improved

Author response:

Following the reviewer’s suggestion, the manuscript underwent proof-reading and language has been improved.

Comment 7:

Here must be more section like limitation of genomic evolutionary process the epilogue, Future prospects

Author response:

We modified the conclusion following the reviewer’s suggestion.

Reviewer 2 Report

Please add more figures and tables to illustrate the novelty. 

Need to extensive editing.

Author Response

Comment 1:

Please add more figures and tables to illustrate the novelty

Author response:

New figures have been added to the manuscript to clarify the concepts illustrated in each chapter and to better represent the novelty of our work.

Reviewer 3 Report

In this review, titled "Genomic Evolutionary Perspective on Long and Healthy Life", the authors take a comprehensive look at genomic instability, including mutation accumulation, telomere shortening, epigenetic changes, and other mechanisms that may contribute to the preservation or damage of genomic information.

Overall, the manuscript is well written in a popular science style. Its content may be of interest to both specialists and people interested in this topic, so I recommend it for publication in IJMS

Minor revision

References must be given in accordance with the rules of the journal

Author Response

Comment 1:

Reference must be given in accordance with the rules of the journal

Author response:

References have been given according to the latest journal guidelines: “Your references may be in any style, provided that you use the consistent formatting throughout. It is essential to include author(s) name(s), journal or book title, article or chapter title (where required), year of publication, volume and issue (where appropriate) and pagination. DOI numbers (Digital Object Identifier) are not mandatory but highly encouraged”. We updated the reference style to Multidisciplinary Digital Publishing Institute style.

Round 2

Reviewer 1 Report

The article by Veschetti et al has now improved alot and can be proceed further for publication. 

minor changes in terms of grammer and phrasing are reqiured